# A Novel Interval Iterative Multi-Thresholding Algorithm Based on Hybrid Spatial Filter and Region Growing for Medical Brain MR Images

Yuncong Feng [1,2,3], Yunfei Liu [1], Zhicheng Liu [1], Wanru Liu [1], Qingan Yao [1,2,*] and Xiaoli Zhang [3,4,*]

1 College of Computer Science and Engineering, Changchun University of Technology, Changchun 130012, China
2 Artificial Intelligence Research Institute, Changchun University of Technology, Changchun 130012, China
3 Key Laboratory of Symbolic Computation and Knowledge Engineering of Ministry of Education, Jilin University, Changchun 130012, China
4 College of Computer Science and Technology, Jilin University, Changchun 130012, China
* Correspondence: yao@ccut.edu.cn (Q.Y.); zhangxiaoli@jlu.edu.cn (X.Z.); Tel.: +86-150-4304-6596 (X.Z.)

**Abstract:** Medical image segmentation is widely used in clinical medicine, and the accuracy of the segmentation algorithm will affect the diagnosis results and treatment plans. However, manual segmentation of medical images requires extensive experience and knowledge, and it is both time-consuming and labor-intensive. To overcome the problems above, we propose a novel interval iterative multi-thresholding segmentation algorithm based on hybrid spatial filter and region growing for medical brain MR images. First, a hybrid spatial filter is designed to perform on the original image, which can make full use of the spatial information while denoising. Second, the interval iterative Otsu method based on region growing is proposed to segment the original image and its filtering layer. The initial thresholds can be quickly obtained by region growing algorithm, which can reduce the time complexity. The interval iterative algorithm is used to optimize the thresholds. Finally, a weighted strategy is used to refine the segmentation results. The segmentation results of our proposed algorithm outperform other comparison algorithms in both subjective and objective evaluations. Subjectively, the obtained segmentation results have clear edges, complete and consistent regions. We use the uniformity measure ($U$) for objective evaluation, and the $U$ value is significantly higher than other comparison algorithms. The proposed algorithm achieved an average $U$ value of 0.9854 across all test images. The proposed algorithm can segment medical images well and expand the doctor's ability to utilize medical images.

**Keywords:** multi-threshold; region growing; interval iteration; medical image segmentation

## 1. Introduction

Image segmentation is one of the most critical image processing techniques. Image segmentation is to divide the target pixels with the same features into the same area and divide the target pixels with different characteristics into different areas [1,2]. Image segmentation performs an important role in medical image processing, and it is widely used in various aspects of the medical field [3], such as lesion localization, quantitative analysis of tissue volume, study of anatomical tissue, and subsequent treatment planning [4,5]. As the cornerstone of image processing, image segmentation has received extensive research and attention from scholars.

In recent years, researchers have conducted a lot of study on image segmentation. Many image segmentation methods have been proposed, including threshold segmentation methods [6], cluster segmentation methods [7], edge detection methods [8], region segmentation methods [9], graph cutting methods [10], methods based on artificial neural network [11–16], etc. The threshold segmentation method has the characteristics of

simplicity and efficiency, and the most classic algorithms include the Otsu method [17], the maximum entropy method [18], and the minimum error method [19]. The Otsu's thresholding method is a well-known method for medical images segmentation [20]. The key to the threshold segmentation method is the selection of the threshold, which directly determines the quality of image segmentation.

The threshold segmentation method is divided into single threshold segmentation and multi-threshold segmentation. Single threshold segmentation is to divide the image into two regions: foreground and background [21]. However, a single threshold segmentation can no longer meet clinical medical needs because of the complexity of medical images. For example, an MR (magnetic resonance) brain image can be divided into five regions in general, including gray matter, white matter, cerebrospinal fluid, skull, and background. Therefore, it is necessary to expand single threshold segmentation to multi-threshold segmentation [22]. When single threshold segmentation is extended to multi-threshold segmentation, it often causes a huge increase in time complexity and a decrease in segmentation accuracy. Therefore, how to propose a fast image segmentation method with high segmentation quality and accuracy is still a challenging task.

To solve the problem of high time complexity, Siyan Liu proposed a method of merging histogram regions to find the thresholds [23]. The optimal solution can be obtained by reducing the histogram region number in iterations. This method reduces the time complexity, but the segmentation accuracy is not high enough. Swarm intelligence algorithms can find optimal solutions in the solution space [24,25]. These methods can overcome the shortcomings of the exhaustive work. Therefore, combining image segmentation algorithms with swarm intelligence algorithms has become a research hotspot. The swarm intelligence algorithms include genetic algorithm (GA), ant colony algorithm (ACA), particle swarm algorithm (PSO), sparrow search algorithm (SSA), wolf swarm algorithm (WSA), Bacterial Foraging Algorithm (BF), adaptive bacterial foraging algorithm (ABF), Real coded Genetic Algorithm (RCGA), nelder–mead simplex method (NMS), etc. [26–28]. These algorithms are classic optimization algorithms. Due to its strong robustness and excellent solving ability, the Otsu method based on the above optimization algorithm has been expanded [29–33].

Accurate segmentation of medical images is a very difficult work problem due to the weaknesses of strong noise, complex tissue structure, blurred area boundaries, etc. To overcome this problem, Liu et al. extended Otsu to become two-dimensional, which fully considers the spatial information of image pixels and their neighborhoods [34]. It can achieve ideal segmentation results even when the original images with low contrast and low signal-to-noise ratio. Wang et al. proposed a 3D Otsu algorithm which comprehensively considers the original image, the neighborhood median and neighborhood mean information of the original image [35]. This method has stronger anti-noise performance and achieves better performance than 2D Otsu. Suhas S et al. proposed a method which combines image neighborhood mean and neighborhood median to suppress noise. It can reduce the noise well in MR images and preserve the structure of medical images detail [36]. Cai et al. proposed an iterative threshold segmentation method, which can segment weak objects and fine details well [37]. Inspired by this method, our team proposed a new interval iteration method and extended it to multiple thresholds [38]. However, the time complexity is too high. At present, there is no algorithm that can adapt to all images and achieve good segmentation results with low time complexity. For medical images, the structure is complex, the shape is irregular, and a lot of noise is easily generated during the acquisition process. In recent years, many scholars have proposed various methods to overcome these shortcomings, but medical image segmentation is still a challenging task [39,40].

To improve the segmentation accuracy of medical images and reduce the time complexity, we propose a novel interval iterative multi-thresholding algorithm based on hybrid spatial filter and region growing. The proposed method fully considers the neighborhood information in original image. It reduces the impact of noise on the quality of image segmentation, and significantly reduces the time complexity. The contributions of this paper can be summarized as follows:

(1)     A hybrid spatial filter is proposed to achieve image multi-scale decomposition which denoising while preserving more details. The proposed filter makes full use of the spatial information in the original image. It can improve the accuracy of image segmentation and make the algorithm more powerful and robust.

(2)     We proposed an interval iterative Otsu method based on region growing (RGIIM). It quickly obtains the growth threshold through the region growing method (RGM) and uses the idea of interval iteration to optimize the thresholds. This method achieves satisfactory segmentation results with minimal time cost.

(3)     A weighted strategy is used to fuse the segmentation result of the original image and its hybrid layers to make the final segmentation result more accurate.

The rest of the paper is organized as follows: Section 2 details the interval iterative Otsu method based on region growing. Section 3 describes our proposed algorithm. Section 4 depicts segmentation results and analysis on MR brain images. Conclusions and future work are presented and discussed in Section 5.

## 2. Interval Iterative Otsu Method Based on Region Growing

### 2.1. Otsu Method

The Otsu method refers to a criterion function which uses the maximum inter-class variance as the threshold selection. Assuming that the size of an image is M × N, the value range of gray level is [0, 255]. The number of thresholds is $K$, $t_i (0 \leq t_i \leq 255, i = 1, \ldots, K)$ represent the thresholds and it is a variable. The image will be divided into $K + 1$ categories, $c_i$ $(i = 1, \ldots, K + 1)$ represent each category. The between-class variance is calculated as follows:

$$\sigma^2(t_1, \ldots, t_K) = \sum_{i=1}^{k+1} \omega_i (\mu_i - \mu_T)^2 \tag{1}$$

where $\mu_i$ represents the average gray level of each type of pixels, and $\mu_T$ represents the average gray level of the entire image. $\omega_i$ represents the class probability of the $i$-th class $(1 \leq i \leq K + 1)$.

We define $n_i (1 \leq i \leq K + 1)$ as the number of pixels whose intensity in the interval of $[t_{i-1}, t_{i-1}]$, and $p_i$ as the probability of pixels in category $c_i$, and $\sum_{i=0}^{255} p_i = 1$. $p_i$, $\mu_i$, $\mu_T$ and $\omega_i$ can be calculated as follows:

$$p_i = \frac{n_i}{M * N} \qquad \mu_i = \sum_{j \in c_i} j p_j \qquad \mu_T = \sum_{j=0}^{255} j p_j \qquad \omega_i = \sum_{j \in c_i} p_j \tag{2}$$

The Otsu threshold segmentation method is to find the optimal thresholds $t_1, \ldots, t_K$ by traversing the entire gray level to maximize the inter-class variance. It can be defined as follows:

$$t_1^*, \ldots, t_K^* = \underset{0 \leq t_1 < \cdots < t_K \leq 255}{\operatorname{argmax}} \{\sigma^2(t_1, \ldots, t_K)\} \tag{3}$$

where $t_1^*, \ldots, t_K^*$ are the optimal thresholds.

### 2.2. Interval Iterative Otsu Method Based on Region Growing

The region growing method (RGM) is to treat each gray level in the image as a small area. By comparing each small area, useless areas are selected to merge with adjacent areas. By continually merging small regions, it stops growing until the number of remaining areas meet our preset conditions. Figure 1 shows the grayscale histogram of an original image and the process of region growing. Figure 1a displays the initial grayscale histogram of image *I*. Figure 1b–d depict the process of region growing. For image *I*, each gray level is regarded as an area. We can divide the image into 256 areas according to the histogram and calculate the information entropy $H(i)$ of the initial 256 areas. It can be defined as follows:

$$H(i) = -p(i) \times \lg(p(i)) \; i = 0, 1, 2, \cdots, 255 \tag{4}$$

where $p(i)$ denotes the probability of gray $i$.

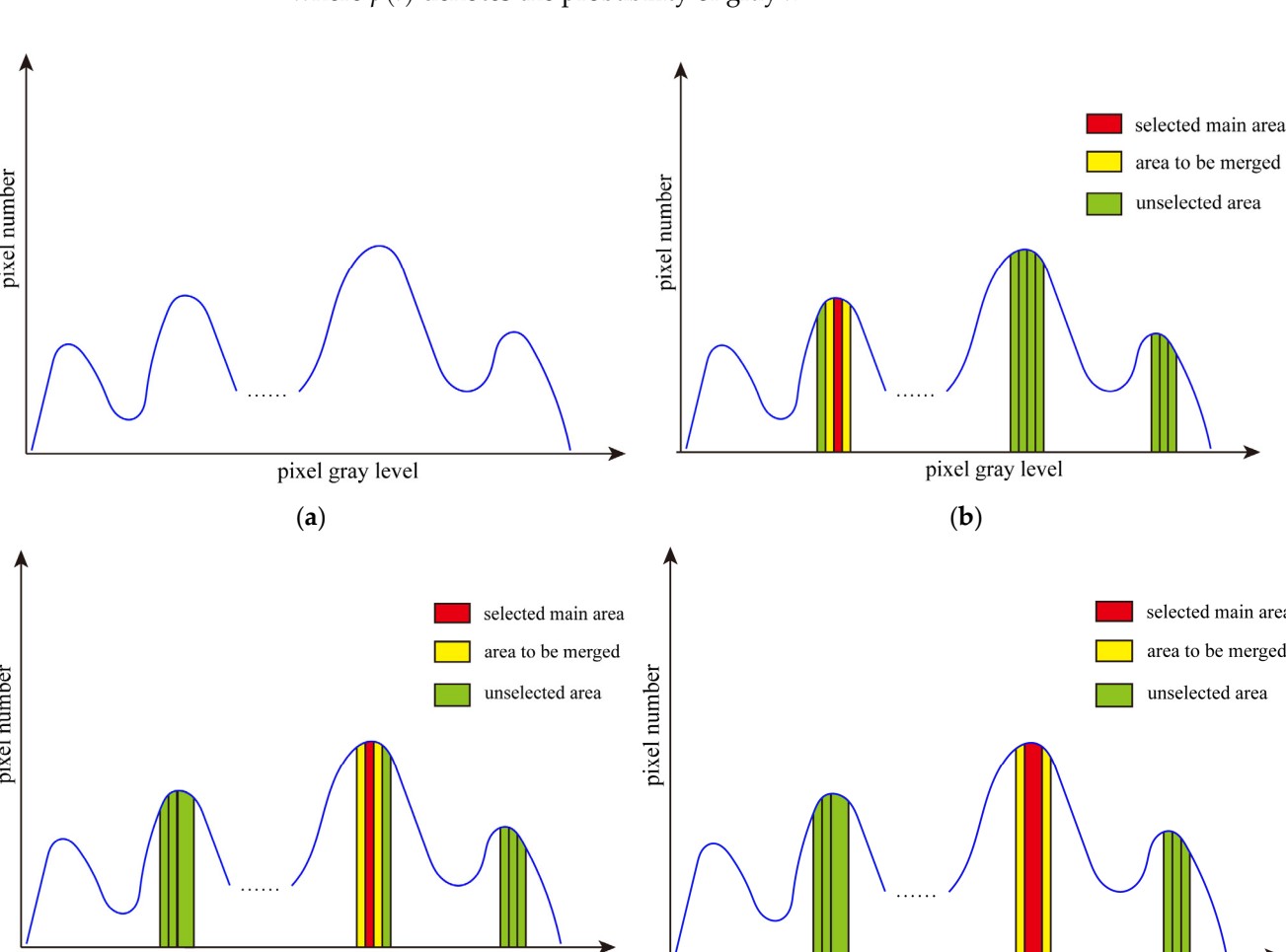

**Figure 1.** The grayscale histogram of the original image and the process of region growing. (**a**) Image grayscale histogram, (**b**) area selection for first region growing, (**c**) results of the first area growth and selection of the main area for the second region growing, and (**d**) the second region growing results and main area selection for the next.

### 2.2.1. The First Stage

In the first stage, the area with the smallest information entropy, $H(i)$, is selected as the main area, which is merged with its adjacent areas. As shown in Figure 1b, the red area is the main area. The yellow areas are to merge into the red one, and the green areas are unselected.

The maximum inter-class variance $W$ of the main area and the area to be merged with are computed separately, and we will select the area with smaller $W$ for merging. The amount of information $D_k$ is taken as the information entropy of the new area, and it will participate in the next main area selection. The calculation of the amount of information $D_k$ as follows:

$$\omega_k = \sum_{i=t_k+1}^{t_{k+1}} P_i \qquad \mu_k = \sum_{i=t_k+1}^{t_{k+1}} \frac{iP_i}{\omega_k} \qquad \sigma_k^2 = \sum_{i=t_k+1}^{t_{k+1}} \frac{(i-\mu_k)^2 P_i}{\omega_k} \qquad D_k = \omega_k \sigma_k^2 \qquad (5)$$

where $P_i$ is probability of the gray level $i$, $\omega_k$ is the probability of each area, $\mu_k$ is the average gray level of each area, $t_k$ is the threshold, and $\sigma_k^2$ is the variance of each area. The definition of the maximum inter-class variance $W$ as follows:

$$\begin{cases} P_0 = \sum_{i=k}^{m} p(i) \\ P_1 = \sum_{i=m}^{n} p(i) \; m < n \end{cases} \begin{cases} u_0 = \sum_{i=k}^{m} i \times p(i) \\ u_1 = \sum_{i=m}^{n} i \times p(i) \; m < n \end{cases} \begin{cases} u = P_0 \times u_0 + P_1 \times u_1 \\ W = P_0 \times (u_0 - u)^2 + P_1 \times (u_1 - u)^2 \end{cases} \tag{6}$$

where $p(i)$ is the probability of gray level $i$, $u$ is the average value of the area, and $W$ is the maximum inter-class variance of the new area.

We take Figure 1c,d as an example to illustrate the process of region growing in the first stage. In Figure 1c, it can be seen that the main area selected for the first time is merged with the area on the right to form a new area. Then, the area with the smallest information entropy $H(i)$ is selected as the main area, which is marked in red. Its two adjacent areas are marked in yellow. The results of the last region growing are shown in Figure 1d. The newly generated area is selected as the main area. The area adjacent to the main area is selected as the area to be merged with. If the main area is an edge area, that is, the main area has only one adjacent area, then it is directly merged with the adjacent area without calculating $W$.

The above operations are repeated. The number of thresholds gradually decreases with region growing, and region growing will stop until it meets our pre-set threshold number. The final thresholds obtained by region growing are shown in Figure 2.

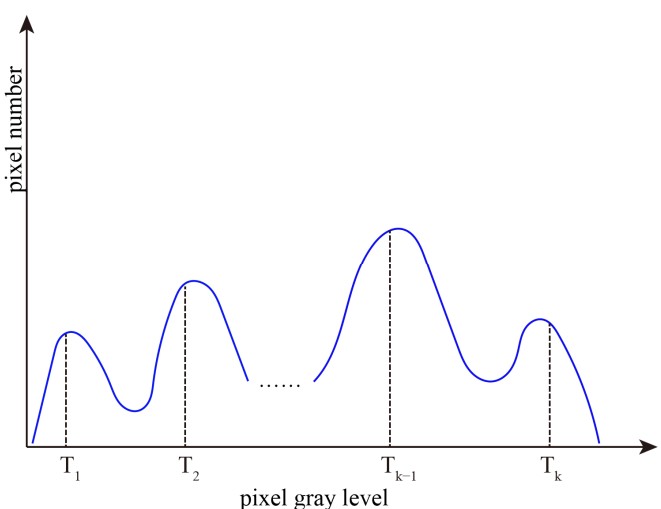

**Figure 2.** Final thresholds obtained by region growing.

2.2.2. The Final Stage

According to the final thresholds $(T_1, T_2, \ldots, T_{k-1}, T_k)$ obtained by region growing methods, the pixels of the original image are divided into $K + 1$ categories, and the mean of each category is recorded as $\mu_{1,I}$ ($i = 1, \ldots, K + 1$). Pixels whose gray value $p_i \leq \mu_{1,1}$ are divided into class $C_1$, while pixels whose gray value $p_i \geq \mu_{1,K+1}$ are divided into class $C_{K+1}$. The remaining pixels are divided into $K$ intervals $[\mu_{1,1}, \mu_{1,2}]$, $[\mu_{1,2}, \mu_{1,3}]$, $\ldots$, $[\mu_{1,K-1}, \mu_{1,K}]$, and $[\mu_{1,K}, \mu_{1,K+1}]$. Figure 3 depicts an example of interval iteration. Figure 3a shows the grayscale interval, and Figure 3b shows the updated grayscale interval after the first interval iteration. In Figure 3a, the intervals filled with different colors are to be iterated continuously next.

Next, Otsu single-threshold segmentation is performed in intervals $[\mu_{1,1}, \mu_{1,2}]$, $[\mu_{1,2}, \mu_{1,3}]$, $\ldots$, $[\mu_{1,K-1}, \mu_{1,K}]$, and $[\mu_{1,K}, \mu_{1,K+1}]$, respectively, to obtain the thresholds $T_{2,i}$ ($i = 1, \ldots, K$) and the corresponding class means $\mu_{2,2i-1}$, $\mu_{2,2i}$ ($i = 1, \ldots, K$). Pixels whose gray value is located in the interval $[\mu_{2,i}, \mu_{2,i+1}]$ ($i = 2, \ldots, 2K - 2$) are divided into $C_j$($j = 2, \ldots, K$), as shown in Figure 3b.

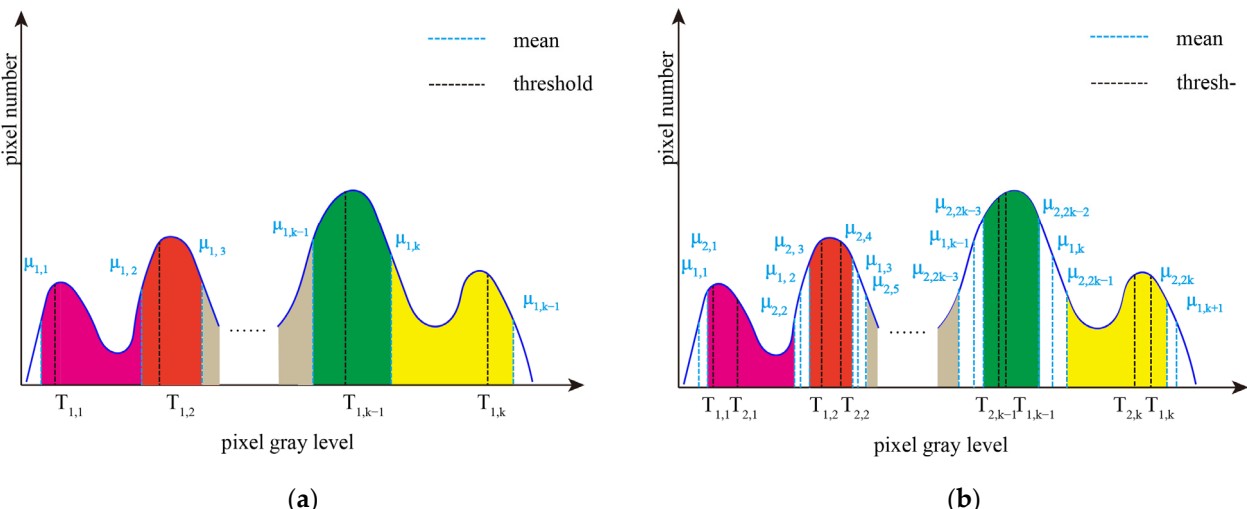

**Figure 3.** An example of interval iteration at different stages. (**a**) Grayscale interval, and (**b**) Updated grayscale interval after interval iteration.

Repeat the above process in class $C_j$ ($j = 2, \ldots, K$) to obtain the threshold and class mean, and re-divide $C_j$ ($j = 2, \ldots, K$) until the threshold is satisfied $\left| T_{h,r} - T_{h-1,r} \right| < \delta$, (where $\delta > 0$). The final obtained $T_{h,r}$ ($r = 1, \ldots, K$) is the optimal threshold $T'_r$ ($r = 1, \ldots, K$). Figure 4 shows all the optimal thresholds obtained by RGIIM.

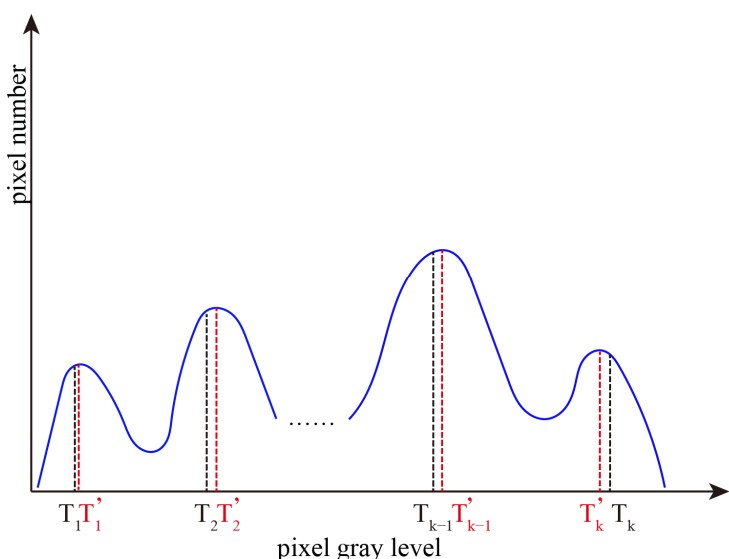

**Figure 4.** The optimal thresholds obtained by interval iterative Otsu algorithm based on region growing.

## 3. The Proposed Algorithm

### 3.1. The Framework

The framework of the proposed algorithm is shown in Figure 5. It can be described as follows:

(1) The original image is processed with the proposed hybrid spatial filter to obtain the hybrid layer.
(2) The proposed RGIIM is executed on the original image and the hybrid layer separately to obtain different sets of segmentation thresholds.
(3) The weighted strategy is performed on the segmentation thresholds to obtained the optimized segmentation thresholds.

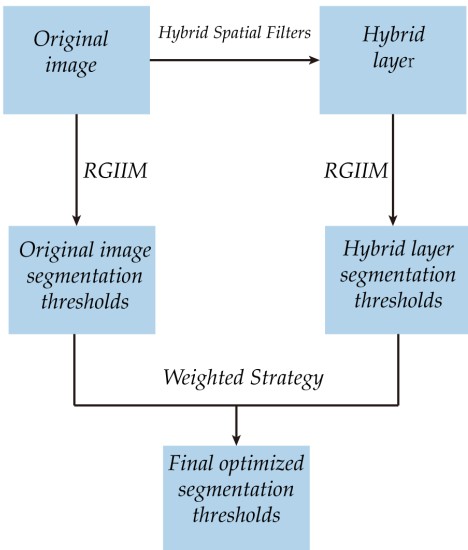

**Figure 5.** Framework of the proposed algorithm.

*3.2. Hybrid Spatial Filter*

In our algorithm, we designed a hybrid spatial filter to improve the accuracy and robustness of image segmentation. The proposed hybrid spatial filter integrates four filters, including Gaussian filter, median filter, mean filter, and mean-median filter. These four filters have been proved by time and experiments, can remove the noise in the image very well, and retain the edge information in the image. It also can make full use of the spatial information of the original image while eliminating noise [30,41]. The common Gaussian filtering, median filter and mean filter are not repeated here. The mean-median filter is defined as follows:

$$mean\text{-}median(x,y) = \frac{\sum\limits_{i=1}^{a*b} \text{average}_{(u,v) \in Sxy}\{g_i(u,v), \text{ median }\}}{a * b} \tag{7}$$

where $Sxy$ represents the set of pixels in the rectangular sliding window with size a∗b, "median" represents the median gray value of pixels in the window centered on the point $(x,y)$, and $g_i(u,v)$ represents the gray value of the *i*-th pixel $(u,v)$ in the window.

We sum up and find the average for the above four filters to form our proposed hybrid spatial filter. Hybrid spatial filter is defined as follows:

$$H(x,y) = \varepsilon G(x,y) + \phi median(x,y) + \gamma mean(x,y) + \eta mean\text{-}median(x,y) \tag{8}$$

where H($x$, $y$) represents the hybrid spatial filter; $G(x, y)$ denotes the Gaussian filter; *median*($x$, $y$) represents the median filter; *mean*($x$, $y$) represents the mean filter; *mean-median*($x$, $y$) represents the mean-median filter; $\varepsilon$, $\phi$, $\gamma$ and $\eta$ are four weights to balance $G(x, y)$, *median*($x$, $y$), *mean*($x$, $y$), *mean-median*($x$, $y$); and $\varepsilon + \phi + \gamma + \eta = 1$. Here, we set $\varepsilon = \phi = \gamma = \eta = 1/4$ empirically.

Figure 6 shows the results of two test images processed by the proposed hybrid spatial filter. It can be seen that the images after hybrid spatial filtering are smoother. The noise has been removed, and the details and structural information in original images are well preserved.

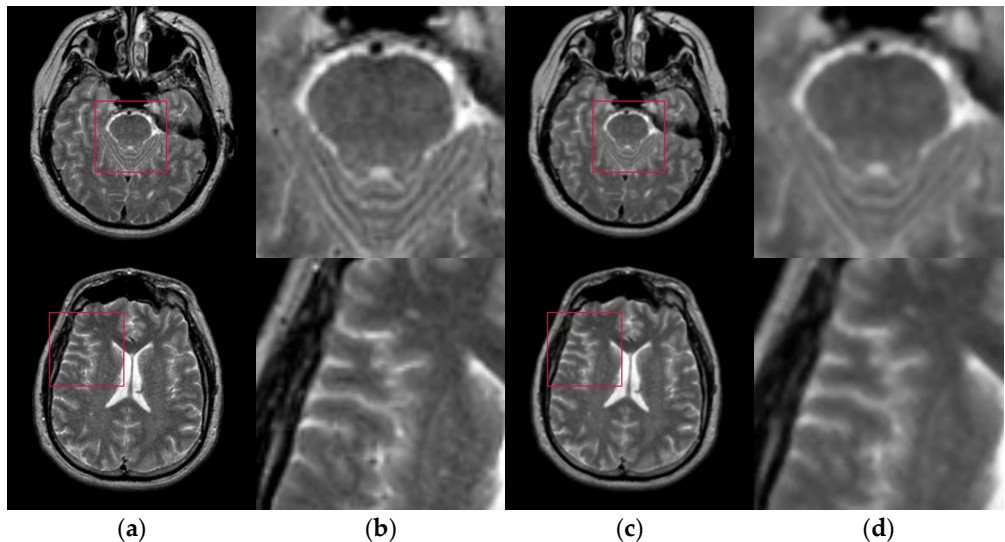

|     (a)     |     (b)     |     (c)     |     (d)     |

**Figure 6.** An example of hybrid spatial filter. (**a**) Original images, (**b**) Part of the enlarged original images (**c**) The results obtained by hybrid spatial filter, (**d**) Part of the enlarged results images.

### 3.3. Weighted Strategy

A weighted strategy is proposed to obtain the final segmentation thresholds for further refinement of segmentation results. The weighted strategy is simple and effective. It is defined as follows:

$$T_f^i = \alpha T_o^i + \beta T_h^i \ (i = 1, \ldots, K) \tag{9}$$

where $T_f^i$ ($i = 1, \ldots, K$) denotes the final optimized segmentation thresholds, $T_o^i$ and $T_h^i$ refer to thresholds of the original image and its hybrid filtering layer, respectively, and $\alpha$ and $\beta$ are two weights to balance $T_o^i$ and $T_h^i$, $\alpha + \beta = 1$. Here, we set $\alpha = \beta = 1/2$ empirically.

## 4. Experimental Results and Analysis

### 4.1. Experimental Protocols

The test images we used in the experiments are from "The Whole Brain Atlas" of Harvard Medical School Image Library (http://www.med.harvard.edu/aanlib/home.html (accessed on 20 September 2021)). Due to limited space, we chose ten brain slices #022~#112 to demonstrate the performance of our proposed algorithm. These ten brain slices are shown in Figure 7. All the experiments in this paper are run on Intel(R) Core (TM) i7-7700HQ CPU @ 2.80 GHz 2.80 GHz, 16 GB RAM, windows 10 and programming language is Python3.6. The parameter settings of the proposed algorithm and the number of thresholds are shown in Table 1.

**Table 1.** Parameter settings of the proposed algorithm.

| Parameter Settings | Description |
| --- | --- |
| $\delta = 0.01$ | Value that stops the iteration for RGIIM |
| $W = 3$ | filter window size |
| $K = 2, 3, 4, 5$ | Number of the thresholds |

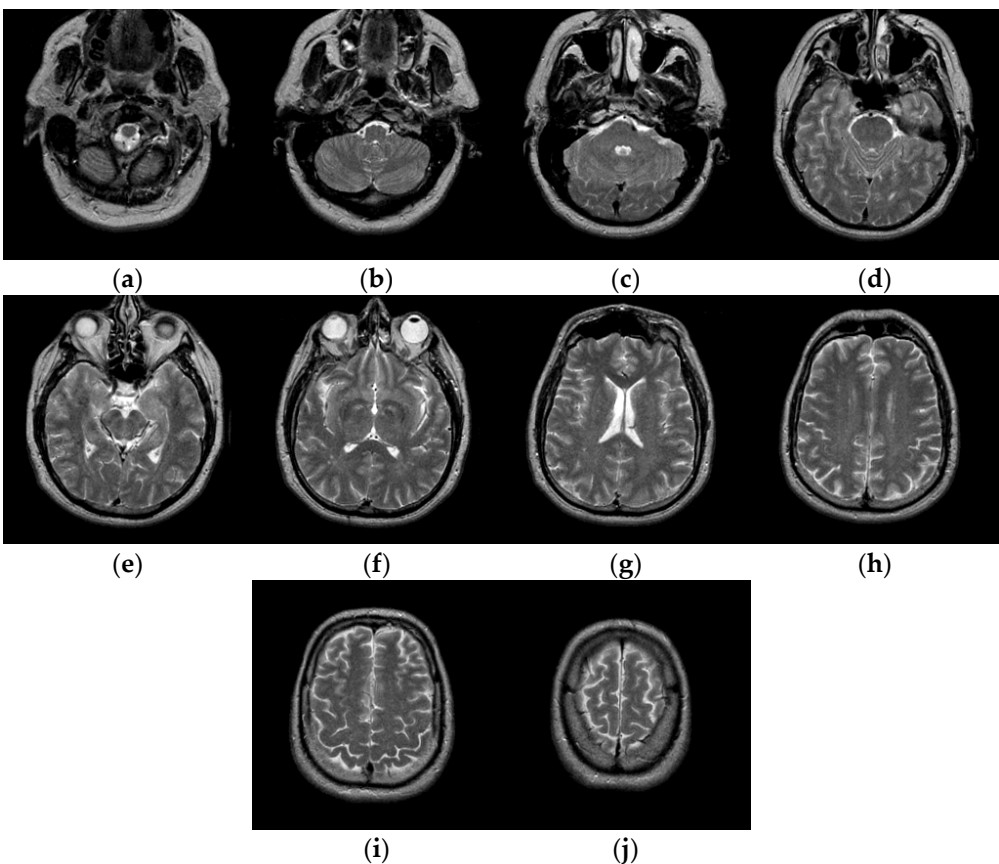

**Figure 7.** Ten brain slices (**a**) slice #022, (**b**) slice #032, (**c**) slice #042, (**d**) slice #052, (**e**) slice #062, (**f**) slice #072, (**g**) slice #082, (**h**) slice #092, (**i**) slice #102, (**j**) slice #112.

### 4.2. Evaluation Measure

We use the uniformity measure as the objective evaluation index in the experiments [42]. The uniformity measure is used to quantitatively evaluate the degree of similarity in different classes divided by the segmentation algorithm. It is defined as follows:

$$U = 1 - 2*n*\frac{\sum\limits_{j=1}^{K}\sum\limits_{i\in R_j}\left(I_i - \mu_{R_j}\right)^2}{Num*(I_{\max} - I_{\min})} \tag{10}$$

where $n$ represents the number of thresholds; $Num$ represents the number of all pixels in the original image $I$; $I_{\max}$ and $I_{\min}$, separately, represent the maximum gray value and minimum gray value of the pixels in the image $I$; $R_j$ denotes the $j_{\text{th}}$ segmented area; $I_i$ represents the gray value of pixel $i$; and $\mu_{R_j}$ represents the grayscale mean of the pixels in the segmented region $R_j$. The value range of $U$ is [0, 1]. The larger the $U$ value, the better the regional uniformity in the segmentation results, and the better the segmentation effect and vice versa.

### 4.3. Comparison between the Proposed Method and Other Methods

To verify the performance of the proposed algorithm, five representative multi-threshold segmentation algorithms are selected for comparative experiments. The five comparative experimental algorithms are: (1) image threshold segmentation algorithm based on particle swarm optimization (PSO), (2) image threshold segmentation algorithm based on bacterial foraging (BF), (3) image threshold segmentation algorithm based on adaptive bacterial foraging (ABF), (4) image threshold segmentation algorithm based on Nelder-Mead simplex (NMS), (5) image multi-threshold segmentation algorithm based

on real coded genetic algorithm (RCGA) [43]. The number of thresholds is set to $K = 2, 3, 4$, and $5$, respectively. Due to the limited space, we only show the segmentation results of the proposed algorithm in this paper. Figure 8 shows the segmentation results of Slice#022~#112 when the threshold $K = 2, 3, 4, 5$. Intuitively, the proposed algorithm can better segment each region of the experimental image, and the continuity of different regions is well guaranteed. At the same time, the visual effects are satisfactory.

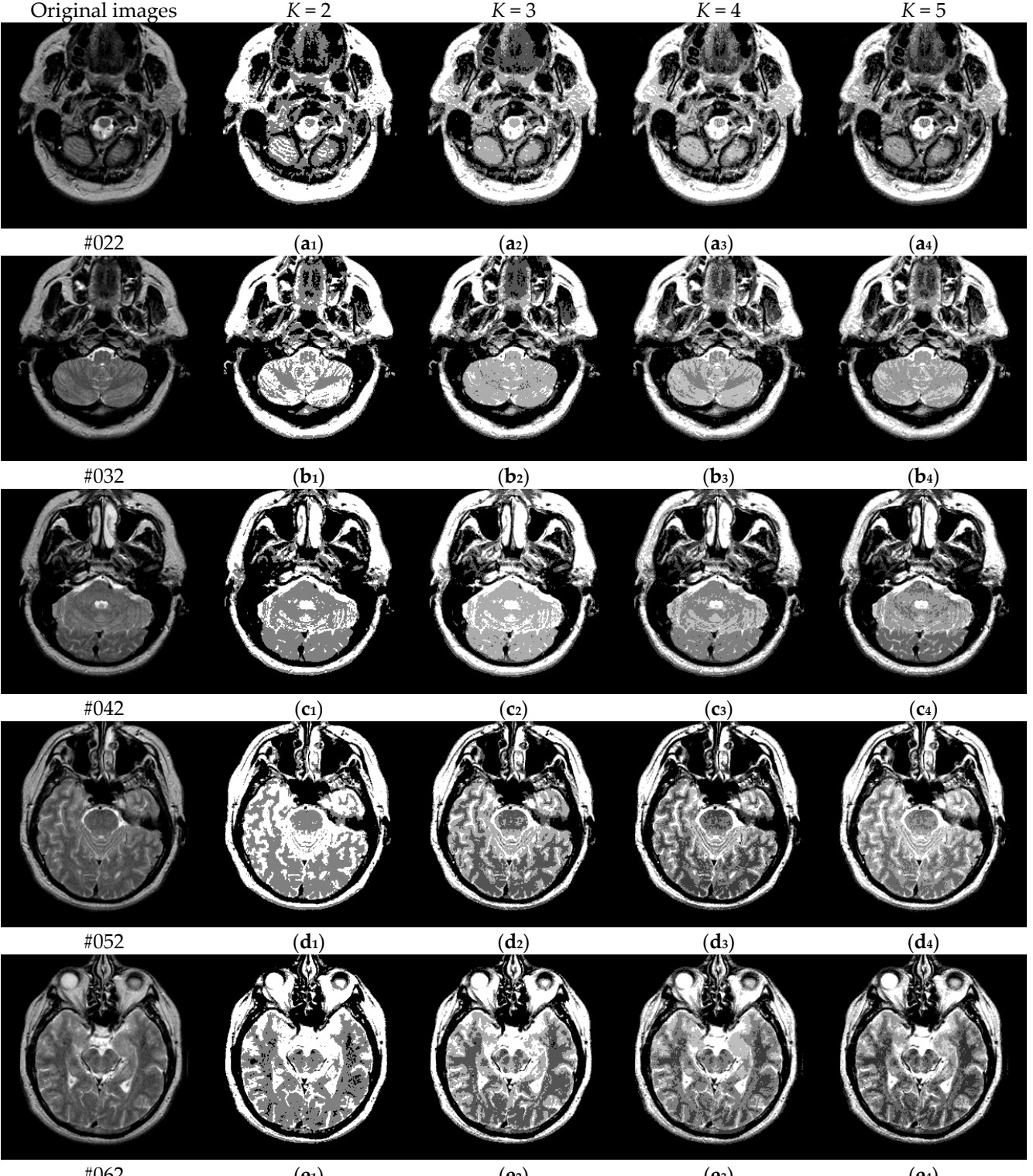

**Figure 8.** *Cont.*

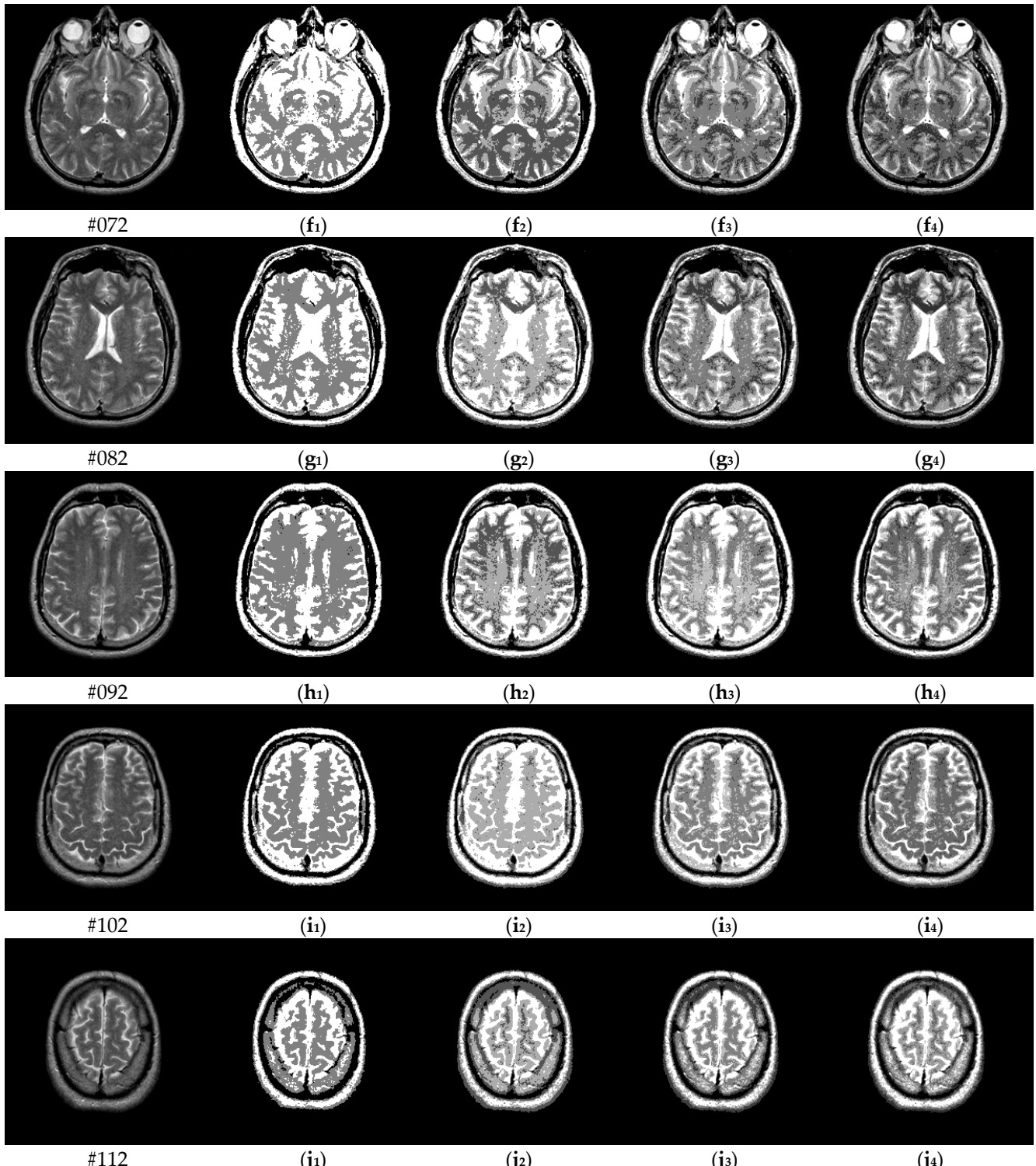

**Figure 8.** Segmentation results obtained by the proposed algorithm for brain slices #022~#112; ($a_1$–$j_1$) display the results of 2-thresholding; ($a_2$–$j_2$) display the results of 3-thresholding; and ($a_3$–$j_3$) display the results of 4-thresholding; ($a_4$–$j_4$) display the results of 5-thresholding.

The uniformity measures ($U$) values of the proposed algorithm and other comparison algorithms with $K = 2, 3, 4, 5$, respectively, are shown in Table 2. The highest $U$ values are marked in bold. In general, our proposed algorithm achieves the best or better evaluation results on all test images. For #042, #052, #062, #082, #112, the proposed algorithm achieves

suboptimal results when $K = 5$. The difference of the objective evaluation $U$ value is small and acceptable. Overall, our proposed algorithm achieves the best segmentation results in test images, outperforming five different contrasting algorithms.

**Table 2.** Comparison of the uniformity measure for different segmentation algorithms.

| Test Images | Number of Thresholds ($K$) | Uniformity Measure ($U$) | | | | | |
|---|---|---|---|---|---|---|---|
| | | **Proposed** | **PSO** | **BF** | **ABF** | **NMS** | **RCGA** |
| #022 | 2 | **0.9870** | 0.9552 | 0.9569 | 0.9569 | 0.9569 | 0.9569 |
| | 3 | **0.9894** | 0.9672 | 0.9708 | 0.9696 | 0.9769 | 0.9769 |
| | 4 | **0.9904** | 0.9420 | 0.9765 | 0.9698 | 0.9824 | 0.9824 |
| | 5 | **0.9917** | 0.9435 | 0.9786 | 0.9785 | 0.9752 | 0.9788 |
| #032 | 2 | **0.9845** | 0.9368 | 0.9342 | 0.9342 | 0.9342 | 0.9342 |
| | 3 | **0.9886** | 0.9619 | 0.9716 | 0.9600 | 0.9796 | 0.9801 |
| | 4 | **0.9891** | 0.9144 | 0.9697 | 0.9766 | 0.9848 | 0.9848 |
| | 5 | **0.9905** | 0.9422 | 0.9668 | 0.9767 | 0.9851 | 0.9843 |
| #042 | 2 | **0.9827** | 0.9271 | 0.9246 | 0.9246 | 0.9246 | 0.9246 |
| | 3 | **0.9812** | 0.9585 | 0.9721 | 0.9689 | 0.9548 | 0.9548 |
| | 4 | **0.9868** | 0.9465 | 0.9752 | 0.9821 | 0.9865 | 0.9865 |
| | 5 | 0.9871 | 0.9348 | 0.9724 | 0.9766 | 0.9845 | **0.9877** |
| #052 | 2 | **0.9837** | 0.9158 | 0.9128 | 0.9128 | 0.9068 | 0.9128 |
| | 3 | **0.9860** | 0.9523 | 0.9713 | 0.9673 | 0.8800 | 0.9467 |
| | 4 | 0.9854 | 0.9372 | 0.9764 | 0.9834 | 0.8982 | **0.9856** |
| | 5 | **0.9880** | 0.9240 | 0.9735 | 0.9782 | 0.9842 | 0.9868 |
| #062 | 2 | **0.9759** | 0.9192 | 0.9047 | 0.9049 | 0.9015 | 0.9015 |
| | 3 | **0.9799** | 0.8777 | 0.9135 | 0.9029 | 0.9030 | 0.9030 |
| | 4 | **0.9849** | 0.9236 | 0.8856 | 0.8988 | 0.8989 | 0.8989 |
| | 5 | 0.9851 | 0.8505 | 0.9527 | 0.9325 | 0.9835 | **0.9855** |
| #072 | 2 | **0.9723** | 0.9068 | 0.9041 | 0.9041 | 0.9041 | 0.9041 |
| | 3 | **0.9788** | 0.9034 | 0.9084 | 0.8985 | 0.8992 | 0.8992 |
| | 4 | **0.9830** | 0.8809 | 0.8876 | 0.8804 | 0.8666 | 0.8666 |
| | 5 | **0.9878** | 0.9531 | 0.8881 | 0.8876 | 0.9818 | 0.9825 |
| #082 | 2 | **0.9782** | 0.9120 | 0.9091 | 0.9091 | 0.9091 | 0.9091 |
| | 3 | **0.9734** | 0.8852 | 0.8621 | 0.8661 | 0.8849 | 0.8849 |
| | 4 | **0.9816** | 0.8619 | 0.8479 | 0.8622 | 0.8695 | 0.8695 |
| | 5 | 0.9847 | 0.9372 | 0.9188 | 0.9105 | 0.9854 | **0.9857** |
| #092 | 2 | **0.9870** | 0.9131 | 0.9156 | 0.9131 | 0.9131 | 0.9131 |
| | 3 | **0.9877** | 0.8607 | 0.8751 | 0.8827 | 0.8786 | 0.8786 |
| | 4 | **0.9863** | 0.9490 | 0.8583 | 0.8514 | 0.8240 | 0.8641 |
| | 5 | **0.9887** | 0.8684 | 0.8923 | 0.8401 | 0.9880 | 0.9876 |
| #102 | 2 | **0.9864** | 0.9383 | 0.9250 | 0.9250 | 0.9250 | 0.9250 |
| | 3 | **0.9851** | 0.8768 | 0.8977 | 0.9097 | 0.9179 | 0.9179 |
| | 4 | **0.9888** | 0.9256 | 0.9410 | 0.9050 | 0.9871 | 0.9871 |
| | 5 | **0.9921** | 0.8446 | 0.9180 | 0.9181 | 0.9907 | 0.9895 |
| #112 | 2 | **0.9875** | 0.9356 | 0.9403 | 0.9404 | 0.9404 | 0.9404 |
| | 3 | **0.9906** | 0.9147 | 0.9666 | 0.9769 | 0.9863 | 0.9890 |
| | 4 | **0.9899** | 0.9751 | 0.9824 | 0.9825 | 0.9885 | 0.9896 |
| | 5 | 0.9887 | 0.9735 | 0.9822 | 0.9830 | **0.9915** | 0.9914 |

Table 3 shows optimal threshold values obtained by different segmentation algorithms on the test images. Additionally, the final thresholds selected by different algorithms are different from each other.

**Table 3.** Comparison of optimal threshold values obtained by applying different segmentation algorithms to the test images.

| Test Images | Number of Thresholds ($K$) | Optimal Threshold Values | | | | | |
| --- | --- | --- | --- | --- | --- | --- | --- |
| | | Proposed | PSO | BF | ABF | NMS | RCGA |
| #022 | 2 | 33, 92 | 97, 184 | 96, 184 | 95, 184 | 96, 184 | 96, 184 |
| | 3 | 17, 56, 110 | 69, 138, 207 | 65, 131, 186 | 69, 114, 185 | 58, 116, 185 | 58, 115, 185 |
| | 4 | 13, 37, 74, 116 | 83, 116, 175, 207 | 52, 99, 148, 186 | 58, 113, 174, 208 | 43, 87, 132, 185 | 44, 87, 131, 186 |
| | 5 | 13, 37, 74, 108, 134 | 76, 119, 154, 184, 214 | 44, 90, 127, 170, 208 | 43, 88, 130, 176, 208 | 44, 104, 140, 176, 214 | 44, 86, 127, 174, 208 |
| #032 | 2 | 32, 90 | 107, 185 | 110, 185 | 110, 185 | 110, 185 | 109, 185 |
| | 3 | 24, 70, 116 | 74, 157, 192 | 72, 120, 198 | 81, 134, 187 | 56, 115, 186 | 53, 116, 185 |
| | 4 | 13, 43, 87, 127 | 95, 125, 164, 194 | 63, 119, 173, 208 | 58, 102, 142, 190 | 39, 83, 132, 189 | 39, 84, 131, 189 |
| | 5 | 11, 37, 67, 94, 125 | 80, 112, 139, 186, 213 | 63, 101, 140, 175, 207 | 52, 87, 128, 167, 198 | 29, 75, 124, 173, 207 | 34, 78, 123, 174, 207 |
| #042 | 2 | 46, 107 | 111, 183 | 114, 184 | 114, 184 | 113, 184 | 114, 183 |
| | 3 | 18, 60, 107 | 80, 148, 178 | 70, 136, 188 | 74, 130, 185 | 84, 132, 188 | 84, 132, 187 |
| | 4 | 18, 58, 100, 140 | 81, 125, 164, 197 | 62, 112, 156, 194 | 50, 100, 143, 190 | 29, 76, 128, 187 | 30, 75, 127, 188 |
| | 5 | 18, 55, 86, 112, 142 | 82, 115, 142, 184, 214 | 58, 114, 151, 188, 218 | 53, 97, 144, 184, 218 | 31, 76, 126, 178, 217 | 25, 69, 114, 156, 194 |
| #052 | 2 | 48, 97 | 119, 186 | 117, 186 | 117, 186 | 118, 185 | 118, 185 |
| | 3 | 44, 87, 122 | 89, 113, 187 | 102, 156, 206 | 107, 158, 204 | 109, 166, 207 | 109, 165, 203 |
| | 4 | 42, 80, 105, 130 | 79, 111, 141, 208 | 93, 124, 171, 210 | 90, 129, 173, 210 | 94, 132, 175, 210 | 91, 131, 174, 209 |
| | 5 | 26, 55, 80, 105, 130 | 65, 85, 131, 162, 203 | 56, 112, 144, 175, 209 | 56, 95, 133, 167, 203 | 20, 67, 120, 167, 207 | 24, 67, 118, 166, 203 |
| #062 | 2 | 66, 109 | 109, 186 | 119, 190 | 119, 186 | 121, 187 | 121, 187 |
| | 3 | 53, 89, 125 | 112, 167, 187 | 97, 133, 183 | 102, 147, 199 | 101, 148, 195 | 101, 147, 196 |
| | 4 | 33, 76, 108, 149 | 85, 134, 180, 203 | 98, 140, 182, 218 | 93, 135, 175, 212 | 94, 134, 176, 211 | 94, 134, 175, 211 |
| | 5 | 33, 76, 98, 125, 156 | 99, 119, 157, 181, 203 | 73, 104, 139, 184, 213 | 79, 111, 145, 179, 212 | 28, 68, 120, 168, 208 | 20, 65, 113, 158, 200 |
| #072 | 2 | 47, 91 | 116, 177 | 117, 179 | 117, 179 | 118, 179 | 117, 179 |
| | 3 | 47, 89, 123 | 96, 178, 207 | 95, 147, 202 | 99, 150, 190 | 100, 142, 188 | 99, 141, 187 |
| | 4 | 25, 73, 106, 145 | 96, 124, 161, 187 | 94, 129, 173, 214 | 95, 134, 174, 214 | 100, 140, 179, 214 | 99, 140, 179, 213 |
| | 5 | 25, 72, 99, 129, 174 | 72, 112, 151, 178, 197 | 87, 109, 139, 178, 210 | 87, 119, 150, 180, 214 | 10, 64, 120, 172, 211 | 14, 64, 119, 171, 211 |
| #082 | 2 | 48, 96 | 110, 170 | 112, 169 | 111, 170 | 112, 169 | 111, 169 |
| | 3 | 20, 72, 100 | 103, 136, 198 | 114, 155, 210 | 111, 155, 201 | 103, 146, 189 | 103, 146, 190 |
| | 4 | 20, 72, 98, 134 | 100, 129, 167, 188 | 103, 139, 175, 214 | 99, 135, 170, 210 | 98, 134, 169, 210 | 98, 133, 169, 210 |
| | 5 | 20, 72, 94, 116, 152 | 78, 105, 151, 180, 201 | 81, 122, 150, 182, 212 | 84, 113, 146, 178, 214 | 14, 62, 115, 168, 210 | 10, 62, 107, 148, 190 |
| #092 | 2 | 58, 98 | 109, 175 | 108, 174 | 109, 174 | 109, 173 | 109, 174 |
| | 3 | 35, 79, 107 | 115, 134, 178 | 107, 144, 209 | 104, 158, 207 | 106, 158, 206 | 105, 158, 206 |
| | 4 | 24, 63, 83, 107 | 77, 107, 149, 194 | 100, 129, 164, 208 | 102, 138, 171, 212 | 112, 152, 186, 220 | 97, 136, 211, 173 |
| | 5 | 24, 63, 82, 101, 121 | 90, 113, 165, 185, 206 | 85, 114, 147, 175, 212 | 96, 128, 158, 186, 216 | 10, 64, 110, 160, 205 | 5, 62, 109, 159, 205 |

**Table 3.** *Cont.*

| Test Images | Number of Thresholds (*K*) | Optimal Threshold Values | | | | | |
|---|---|---|---|---|---|---|---|
| | | Proposed | PSO | BF | ABF | NMS | RCGA |
| #102 | 2 | 55, 97 | 98, 166 | 108, 174 | 108, 174 | 108, 173 | 107, 174 |
| | 3 | 25, 64, 97 | 113, 145, 180 | 103, 148, 189 | 98, 146, 189 | 94, 142, 189 | 94, 142, 190 |
| | 4 | 25, 62, 86, 118 | 84, 124, 165, 189 | 79, 122, 164, 200 | 90, 127, 164, 198 | 2, 64, 119, 173 | 1, 63, 120, 174 |
| | 5 | 25, 62, 86, 112, 140 | 99, 128, 147, 194, 218 | 81, 113, 147, 187, 220 | 82, 114, 148, 184, 218 | 9, 62, 106, 147, 190 | 1, 62, 104, 145, 189 |
| #112 | 2 | 54, 100 | 109, 162 | 105, 165 | 105, 164 | 106, 163 | 106, 163 |
| | 3 | 29, 76, 111 | 104, 163, 216 | 79, 134, 180 | 71, 123, 175 | 3, 49, 145 | 1, 70, 142 |
| | 4 | 25, 61, 90, 111 | 63, 130, 153, 206 | 54, 117, 156, 192 | 58, 105, 146, 182 | 4, 63, 132, 178 | 1, 65, 123, 172 |
| | 5 | 18, 49, 76, 94, 111 | 58, 128, 155, 187, 213 | 48, 112, 137, 161, 200 | 47, 108, 142, 171, 197 | 2, 44, 79, 131, 175 | 1, 49, 95, 139, 183 |

### 4.4. Ablation Experiment

To verify the necessity of each step of the proposed algorithm, we conduct ablation experiments. Non-F-RGM means that the image is segmented directly by RGM and without being processed by the hybrid spatial filter. F-RGM means that RGM is only performed on the original image and the filtering layer. Non-F-RGIIM means that the original image is directly segmented using the RGIIM. Proposed represents our proposed complete algorithm. We perform ablation experiments on the 10 test images described in Figure 7. The number of thresholds is set to $K$ = 2, 3, 4, 5, respectively.

Table 4 shows the results of the ablation experiments. The best results are marked in bold. Comparing with Non-F-RGM, the average $U$ value of F-RGM is larger. It means that our proposed hybrid spatial filter has significantly influence on the improvement of segmentation results. Comparing with Non-F-RGM, the average $U$ value of Non-F-RGIIM is improved more significantly. It indicates that the proposed RGIIM is equally important for the improvement of segmentation results, and the idea of optimizing the threshold with interval iteration is effective. Unsurprisingly, the proposed algorithm has achieved the highest average $U$ value. Compared with Non-F-RGIIM, the improvement brought by the hybrid spatial filter is significant especially when $K$ = 2, 5. The improvement is not obvious when $K$ = 3, 4, but the result is also acceptable. In summary, every step of our proposed algorithm is indispensable for medical image segmentation.

**Table 4.** Ablation experimental results.

| Number of Thresholds ($K$) | Average Uniformity Measure ($U$) | | | |
|:---:|:---:|:---:|:---:|:---:|
| | **Non-F-RGM** | **F-RGM** | **Non-F-RGIIM** | **Proposed** |
| 2 | 0.9696 | 0.9729 | 0.9795 | **0.9825** |
| 3 | 0.9723 | 0.9775 | 0.9838 | **0.9840** |
| 4 | 0.9677 | 0.9820 | 0.9864 | **0.9866** |
| 5 | 0.9791 | 0.9850 | 0.9877 | **0.9884** |

### 4.5. Time Complexity Analysis

4.5.1. Proposed Method Time Complexity Analysis

The time consumption of our proposed algorithm includes three parts: first, the original image is processed by the hybrid spatial filter, and the time complexity is $O(L^2 * W)$ level (W is the filter window size). Second, the multi-thresholding based on region growing is performed, and the time complexity is $O(L)$ level. Finally, the time complexity of multi-threshold segmentation based on interval iteration is $O(L)$ level. In sum, the time complexity of this algorithm is $O(L^2 * W)$ level. When the number of thresholds $K$ = 2, 3, 4, 5, respectively, the average running time of the algorithm on 10 test images is shown in Figure 9.

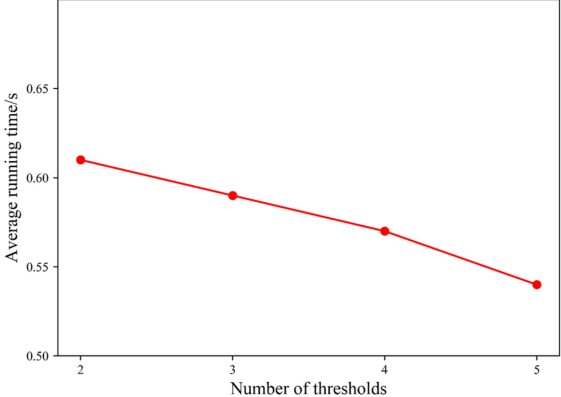

**Figure 9.** The average running time of the proposed algorithm under different thresholds.

In Figure 9, it can be noted that the running time of the proposed algorithm gradually decreases with the increase in the number of thresholds. The overall running time of the algorithm is about 0.6 s, which can be fully applied in real-time systems.

### 4.5.2. Computation Time Comparison

In this section, we compare the computation time of the proposed method with other methods (PSO, BF, ABF, NMS, and RCGA). In Table 5, the time computation of our proposed method for 1 K (1024) pixels is overwhelming compared to other methods. For the comparison methods, PSO takes the least time, which is 0.1057 S. However, our proposed method only took 0.008 s, which is a huge gap. It fully proves that the time cost of our proposed method is tiny.

**Table 5.** Computation time compare with other methods.

| Methods | Proposed | PSO | BF | ABF | NMS | RCGA |
|---------|----------|--------|--------|--------|--------|--------|
| 1 K pixel/s | 0.008 | 0.1057 | 0.2772 | 0.3254 | 0.3548 | 0.2815 |

### 5. Conclusions

In this paper, we proposed a novel interval iterative multi-thresholding segmentation algorithm based on hybrid spatial filter and region growing for medical brain MR images. Experiments show that the hybrid spatial filter can eliminate the noise in the image well while preserve the edge information of the image. It makes the segmentation effect more accurate. The algorithm makes full use of the spatial information in the original image. It quickly obtains initial thresholds by region growing, then the idea of interval iteration is adopted to optimize the thresholds. Finally, a weighting strategy is used to achieve medical image segmentation. Compared with other similar algorithms, our algorithm achieves satisfactory results. Subjectively, good visual effects are achieved, and the segmentation results have good consistency. Objectively speaking, the evaluation metric of our algorithm achieves the best results among all the test images and has a strong anti-noise performance. The time complexity of the algorithm is $O(L^2 * W)$ level. Compared with the comparative algorithm PSO with the least time consumption, for processing 1 K (1024) pixels, the time consumption is about 1/10 of it. This significantly improved the running speed, which can be applied in the real-time system of clinical medicine. We achieve excellent segmentation accuracy with minimal time cost.

**Author Contributions:** Conceptualization, Y.F. and Y.L.; form analysis, Y.F. and Y.L.; methodology, Y.F. and Y.L.; software, Y.L. and Y.F.; visualization, Y.F.; writing—original draft, Y.F. and Y.L.; investigation, Y.F. and X.Z.; validation, Y.F., Z.L. and W.L.; writing—review and editing, Y.F., X.Z. and Q.Y.; data curation, Y.L., Z.L. and W.L.; supervision, Y.F., X.Z. and Q.Y.; project administration, Y.F. and X.Z.; funding acquisition, Y.F. All authors have read and agreed to the published version of the manuscript.

**Funding:** The work was supported by Natural Science Foundation of Jilin Province (NO.YDZJ202201Z YTS422), Youth Growth Science and Technology Plan Project of Jilin Provincial Department of Science and Technology (NO.20210508039RQ), "Thirteenth Five-Year Plan" Scientific Research Planning Project of Education Department of Jilin Province (NO.JJKH20210752KJ, NO.JJKH20200677KJ), Fundamental Research Funds for the Central Universities JLU (NO.93K172020K05), and National Natural Science Foundation of China (NO.61806024).

**Institutional Review Board Statement:** Not applicable.

**Informed Consent Statement:** Not applicable.

**Data Availability Statement:** The data described in this article can be freely and openly accessed at "The Whole Brain Atlas" of Harvard Medical School Image Library (http://www.med.harvard.edu/aanlib/home.html (accessed on 20 September 2021)).

**Acknowledgments:** The authors would like to thank http://www.med.harvard.edu/aanlib/home.html (accessed on 20 September 2021) for providing source medical images.

**Conflicts of Interest:** The authors declare no conflict of interest.

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
