# Peer review of "A Novel Interval Iterative Multi-Thresholding Algorithm Based on Hybrid Spatial Filter and Region Growing for Medical Brain MR Images"

_applsci, doi:10.3390/app13021087_

Round 1
Reviewer 1 Report
In this paper, a novel interval iterative multi-thresholding algorithm based on hybrid spatial filter and region growing for medical brain MR images is proposed. The authors depicted the processing procedures of the proposed method and made some simulations to verify the feasibility of the proposed method. Experiments show that the hybrid spatial filter can eliminate the noise in the image well while preserve the edge information of the image. It makes the segmentation effect more accurate. The paper presents are interesting idea and can be accepted for publication.
Reviewer 2 Report
I attched the comment file.

Reviewer 3 Report
My comments are listed as follows.
(1) The content of section 2.1 needs to be reduced.
(2) Please add more information into Fig.1, such as x-label.
(3) For some case, sub-peaks exists in curves. There are a part of peaks are useless. How to select these peaks used in your method.
(4) The advantages of the research results are not intuitive and not significant. The author needs to highlight the highlights of the research results.
(5) The following papers are related with this work.
[1] Z. Shi, X. Zhou, Spatio-Temporal Dynamic Fields Estimating and Modeling of Missing Points in Data Sets Using a Flexible State-Space Model, APPLIED SCIENCES, 11(19), 9050, 2021.
[2] Q. Zuo, Y. Geng, C. Shen, J. Tan, S. Liu, Z. Liu, Accurate angle estimation based on moment for multirotation computation imaging, Applied Optics, 59(2), 492-499, 2020.
Round 2
Reviewer 2 Report
The authors modified the manuscript appropriately in the reviewer's comment. I have no more comments or suggestions.